# Reliability of a Multidisciplinary Multiparametric Approach in the Surgical Planning of Laryngeal Squamous Cell Carcinomas: A Retrospective Observational Study

**DOI:** 10.3390/jpm12101585

**Published:** 2022-09-26

**Authors:** Davide Rizzo, Claudia Crescio, Pierangela Tramaloni, Laura M. De Luca, Nicola Turra, Alessandra Manca, Paola Crivelli, Chiara R. Tiana, Alberto Fara, Antonio Cossu, Stefano Profili, Mariano Scaglione, Francesco Bussu

**Affiliations:** 1Department of Medical, Surgical and Experimental Science, University of Sassari, 07100 Sassari, Italy; 2Division of Otolaryngology, Azienda Ospedaliero Universitaria, 07100 Sassari, Italy; 3Residency Program in Otolaryngology, University of Cagliari, 09124 Cagliari, Italy; 4Institute of Pathology, Azienda Ospedaliero Universitaria, 07100 Sassari, Italy; 5Institute of Radiology, Azienda Ospedaliero Universitaria, 07100 Sassari, Italy

**Keywords:** laryngeal cancer, squamous cell carcinoma, head and neck, radiological assessment, fibro endoscopy, laryngectomy, histopathology, tumor board, open partial horizontal laryngectomy, OPHL, endoscopic supraglottic laryngectomy, eHSL, total laryngectomy

## Abstract

(1) Background: Endoscopy and morphological imaging are the mainstay of the diagnostic work up of laryngeal squamous cell carcinomas (LSCCs), which can be integrated in a multidisciplinary discussion to obtain a shared pretreatment staging. (2) Methods: A retrospective evaluation of patients, managed at a tertiary university hospital in Italy and submitted to major laryngeal surgery, has been performed. Four different stagings have been defined and compared: epTN (based on endoscopy and physical ENT examination); radTN (based on CT scan); cTN (based on multidisciplinary integration of the two above); pTN based on pathology on surgical samples. Oncological outcomes have been assessed. (3) Results: Three-year relapse free and disease specific survival were 88% and 92.5%, respectively, without significant differences between partial surgeries (*n* = 13) and total laryngectomies (*n* = 32). As for the pretreatment staging, and in particular the T classification, the cTN has been revealed as more reliable than epTN and radTN alone in predicting the final pT (Cohen kappa coefficient: 0.7 for cT, 0.44 for radT, 0.32 for epT). In the partial surgery group, we did not record any positive margin nor local recurrence, with a 100% overall and disease-specific survival. (4) Conclusions: The multidisciplinary approach is fundamental in the definition of the primary lesion in LSCC, in particular in order to safely perform surgical preservation of laryngeal function, which is associated with a higher laryngectomy-free survival than irradiation but to a lower salvageability in case of recurrence.

## 1. Introduction

Laryngeal cancer, mostly squamous cell carcinomas (LSCC), represents one-third of all head and neck cancers, which are the sixth most common cancer worldwide, and can arise in different subsites of larynx, with specific clinical features and therapeutic options [1].

A certain decrease in incidence has been recorded in Western countries, proportional to the success of antismoking campaigns [2,3], which has been more evident in the US, where the incidence of laryngeal cancer has been 2.8/100,000 in 2020, with a constant decrease over the last 30 years. The death rate is around 0.9/100,000 [4].

However, LSCC remains the only human malignancy with uterine body adenocarcinoma without a clear prognostic improvement over the past 30 years [5].

This can derive at least in part from the increasing attention to quality-of-life issues and functional outcomes, which, in the case of larynx, mean speech and swallowing. In fact, in the last 30 years, functional outcomes in LSCC have become almost as important as the fundamental oncological endpoints (survival, relapse) and have produced pressure towards function-preserving therapeutic strategies, that is, the attempt to avoid primary total laryngectomy (the mainstay of laryngeal oncology).

Data from non-surgical organ preservation strategies raised great enthusiasm in the 1990s as they showed that LSCC (with the exclusion of cT4 cases) was highly curable by non-surgical preservation protocols, thus anatomically preserving the larynx [6,7,8,9,10,11].

However, such anatomical preservation through non-surgical strategies often does not correspond to a satisfying functional preservation [12]. In addition, the long-term oncological results of cT2 and cT3 cases are less outstanding than initially estimated [6,7,8,9,10,11,13], with an increased rate of non-cancer-related deaths in patients treated by chemoradiotherapy [14] and laryngectomy after non-surgical preservation failures is notoriously affected by a higher rate of complications and worse short and long-term outcomes [15,16].

Such considerations, combined with emerging data mostly from European groups, describing a markedly higher total laryngectomy-free survival associated with open partial horizontal laryngectomies (OPHL), and endoscopic horizontal supraglottic laryngectomies (eHSL), when indicated [17,18,19,20], contributed to an increasing popularity of surgical functional preservation.

Nevertheless, an inappropriate indication to partial surgery—most often deriving from inappropriate presurgical assessment—is associated with the almost constant need for larynx sacrifice and much higher mortality than non-surgical approaches, as post-surgical recurrences have a definitively lower salvageability [17,18].

Therefore, pretreatment staging, which is of utmost importance in oncology in general, is definitely the key for success in laryngeal oncology in the era of function preservation. Endoscopy is the gold standard for the assessment of mucosal surfaces, but it has not been shown to be reliable for the evaluation of submucosal and paraglottic spaces [21] and, most of all, of cartilages and extralaryngeal deep spaces, which are fundamental parameters both for staging and surgical planning. Hence, morphological imaging is also mandatory (CT scan and/or MRI) for preoperative assessment [21,22].

Multidisciplinary management in tumor boards has become a standard in oncology, and has been demonstrated to significantly improve prognosis in head and neck oncology [23,24]. Tumor boards are also meant to integrate radiological, clinical, and endoscopic data for obtaining the most reliable staging and, therefore, treatment planning.

The aim of the present study is to assess and compare the effectiveness of physical examination/endoscopy (fibrolaryngoscopy), contrast enhanced CT scan, and multidisciplinary evaluation during tumor board meetings based on the integration of the above information in presurgical assessment of laryngeal squamous cell carcinoma (LSCC).

## 2. Materials and Methods

This is a retrospective observational cohort study based on the retrieval of clinical, oncological, and radiological data and of tumor board meeting reports of patients affected by primary and recurrent laryngeal and hypopharyngeal SCCs and treated at the ENT division of AOU Sassari, a tertiary University Hospital in Sardinia, Italy, between August 2017 and December 2020 through a major laryngeal surgery (partial horizontal laryngectomy or total laryngectomy). M+ patients as well as non-surgical cases and cases treated through minor laryngeal surgery (cordectomy for T1-2 glottic primaries) have been excluded from the present analysis. Staging workup has been performed according to the NCCN guidelines for the cancers of the larynx [25] which also included a neck and chest CT scan with iodine i.v. contrast and thin-angled cuts through the larynx, a complete head and neck exam with mirror and fiberoptic examination, and an evaluation of pulmonary function.

### 2.1. Tumor Board Discussion and Treatment Selection

Every head and neck cancer patient in our institution is evaluated at a multidisciplinary tumor board, which completes and defines the clinical staging and gives treatment recommendations, as described also by other groups [26]. The cTNM always results from a multidisciplinary discussion involving different expertises (e.g., head and neck surgeons, radiologists, radiation oncologists, medical oncologists, histopathologists) and using all the clinical as well as radiological data available. In the tumor board, choice of treatment is always based on three orders of parameters:The staging according to American Joint Committee on Cancer (AJCC);Patterns of local spread not included in the AJCC T staging (such as the posterior paraglottic space involvement) as previously described [17,18,20];Patient-related parameters such as age, comorbidities, and preferences.

Namely, with the intent of organ preservation, in most cT3 cases, we suggested a horizontal laryngectomy (type II OPHL or eHSL) when feasible according to previously described criteria [17,20]. Alternatively, radiochemotherapy has been recommended, when indicated, with the intent of organ preservation (cases not included in the present series) [17]. In cT3 cases with a low compliance to surgical or non-surgical organ preservation protocols as a result of comorbidity, social factors, or personal preference, we performed a total laryngectomy. For patients with cT4a SCCs, we usually recommend a total laryngectomy, with the exception of cT4N0 cases with exclusive involvement of the anterior commissure and/or of the anterior part of the thyroid laminae in the absence of extensive extralaryngeal invasion; in these cases, a CHP was sometimes recommended.

All patients underwent bilateral neck dissection, elective/selective (levels II, III, IV) in cN0 cases, comprehensive (levels II-VI) in cN+.

Follow-up visits were scheduled every 3 months in the first year, every 4 months in the second, every 6 months in the third and fourth, and every year from the fifth on and included ceCT (including the chest once a year), endoscopy, and complete head and neck physical examination in order to monitor relapses and detect second primary tumors early.

### 2.2. Data Collection

The present work is substantially an audit aimed at evaluating the effectiveness, strengths, and the weaknesses of a multidisciplinary team in the surgical planning for laryngeal cancers.

Several parameters (see results section), have been collected and analyzed as assessed by history collection, endoscopy and physical examination, follow-up visits, radiology, multidisciplinary discussion, and histopathology. More in detail, parameters concerning the pattern of spread of primary tumor, as assessed by endoscopy, ceCTscan and pathology on the surgical sample, have been recorded and compared (see results section).

Four different stagings have been defined and compared (see results section):epTN (based on endoscopy and physical ENT examination);radTN based on CT scan;cTN (based on multidisciplinary discussion);pTN based on pathology on surgical sample.

### 2.3. Statistical Analysis

Statistical analysis was performed using the JMP in software, release 7.0.1, from the SAS Institute (Cary, NC, USA).

Survival has been evaluated using Kaplan–Meyer curves, which were compared by Log-rank test.

The agreement between epTN, radTN, and cTN and the reference pTN, and therefore their reliability, has been determined by calculating the Cohen kappa coefficient, a measure that takes a value of zero if there is no correlation between the different assessments. To score the significance, the kappa values are ranked. Values less than 0.4, between 0.4 and 0.75, or higher than 0.75 represent poor, fair-to-good, or excellent correlations, respectively.

## 3. Results

Forty-five consecutive patients have been included.

Descriptive statistics are shown in Table 1.

In total, 41 out of 45 patients were operated on for a primary tumor and 4 for recurrences; among these, two had been treated primarily through non-surgical modality and two through surgery.

Twelve patients underwent type-2 OPHL; one underwent EHSL; the remaining 32 patients underwent total laryngectomy. Four recurrences were recorded in the total laryngectomy population (1 local; 1 regional, level VI; 2 distant). One recurrence was recorded in the partial laryngectomy population (regional at level VII).

The exclusion of patients treated after 2020 allowed a reasonable and reliable follow-up in order to assess oncological outcome (average follow-up, 24 months).

Relapse-free survival (RFS, Figure 1A), overall (OS, Figure 1B), and disease-specific (DSS, Figure 1C) survival were 88%, 74%, and 92.5%. respectively, at 3 years (Figure 1).

When comparing patients submitted to partial laryngeal surgery (*n* = 13) with those submitted to total laryngectomy (*n* = 32), no differences are found as for 3-year RFS and DSS, while partial laryngectomies are associated with a significantly longer OS (Figure 2C, *p* = 0.04 at Log-rank test). This finding is expected because partial laryngectomies are indicated in patients with no neck nodes and no massive extralaryngeal extension (usually lower stages), less comorbidities (in particular CPOD), and younger age.

The results concerning the primary endpoints of the present study are shown in Table 2. The ability to predict the pTN with the different pretreatment stagings (epTN, radTN, cTN) as estimated by the calculation of kappa value is always fair to good for radTN and cTN, while it remains poor for epTN, both in the assessment of T and N classification. However, interestingly, the ability to predict the pathological T classification of LSCC, which is key for the success of surgery, and in particular of partial surgery, is increased after discussion and integration of endoscopy/physical exam (epTN) and radiology (radTN) in the multidisciplinary tumor board (cTN, kappa 0.7).

## 4. Discussion

The careful and extensive data collection and the long follow-up compensate for the relative smallness of the sample and allowed to obtain clear and interesting results. The “real world” setting, with only contrast-enhanced CT scan and endoscopy available for staging and definition of the primary lesion and of N, allows one to appreciate the real impact of multidisciplinary management in daily clinical practice.

In fact, the present work clearly confirms that integration of clinical, endoscopic, and radiological data in the multidisciplinary discussion of tumor boards drastically increases the reliability of the pretreatment definition of T classification and the definition of anatomical pattern of spread of the primary lesion in LSCCs, as demonstrated by the higher kappa value and by the extremely low rate of local recurrence (0 in the partial surgery group). Even if the necessity of multidisciplinary management of head and neck cancers has been clearly and extensively proven, its utility for this specific aim in laryngeal oncology had never been previously evaluated.

The indication to partial laryngeal surgery, that is, surgical functional preservation, takes into account patient-related parameters and tumor-related parameters [17,18,27]. Information related to the spread of the primary lesion, partly directly contributing to T classification, and partly not, are critical for an oncologically safe indication to partial laryngeal surgery both in primary and in recurrent LSCC. We previously reported that a careful workup and rigid criteria for the recommendation of OPHL lead to similar disease-specific but markedly higher laryngectomy-free survival than non-surgical preservation protocols (radiochemotherapy) [17]. On the other hand, it is common experience that inappropriate indications to partial laryngeal surgery and OPHL in particular, resulting in positive resection margins, not only compromise organ preservation but also, most importantly, disease-specific survival [28,29,30,31,32]. Therefore, one can affirm that a multidisciplinary shared definition of the primary tumor is mandatory to perform surgical preservation in laryngeal oncology [33]. In the present series, we recorded only one positive margin and subsequent local recurrence in a total laryngectomy performed for a post-surgical recurrence, and no positive margins nor local recurrences in the partial surgery group (with a 100% overall and disease-specific survival), confirming the effectiveness of a methodical multidisciplinary assessment of the primary lesion in avoiding local failures in head and neck oncology and its decisive impact on disease specific survival. Notably, in the partial surgery group of the present series, cT and cN classification corresponded perfectly to pT and pN classification, and this was the main key for the clear oncological success.

The markedly higher overall survival in partial surgery patients in the present series is of course influenced by selection biases, that is, patient-related parameters, and mainly more severe comorbidities, in particular in the lung, which contraindicate partial surgeries.

As for N, the added value of multidisciplinary integration of clinical and radiological data is definitely less beneficial, as one mostly relyinf almost exclusively on radiological findings kappa value of radN is the same as cN as obtained through the multidisciplinary discussion. As a consequence, the prediction by the tumor board of pN is less reliable than that of pT, with potential issues in patients submitted to OPHL in case of histopathological node involvement, which may be an indication for an adjuvant radio+/-chemotherapy, potentially compromising functional results [17,34,35]. This finding, on one hand, confirms the appropriateness of the attitude to always perform bilateral neck dissection in advanced LSCC and, on the other hand, suggests that tools allowing an increase in the sensitivity and specificity in the diagnosis of nodal metastasis should be considered in the work up, including ultrasound-guided fine needle aspiration cytology (FNAC) [36] and PET-CT, whose interpretation pitfalls should be carefully taken into account anyway [36,37,38,39]. However, in the present series, when only presence vs. absence of regional metastasis is considered, which is the decisive parameter in indicating adjuvant irradiation, the reliability of radTN and of cTN increased, and none of the partial surgeries had indication to postoperative treatment because of unexpected nodal spread (nor because of positive margins).

In the definition of the primary lesion, in order to safely recommend partial surgery, some features need to be accurately excluded independently from T classification, and in particular true arytenoid fixation, involvement of the base of the tongue, massive pre-epiglottic space or vallecular invasion, posterior lateral paraglottic space invasion, interarytenoid diffusion, extensive thyroid cartilage invasion, significant subglottic involvement for type II OPHL (in general down to the level of the superior border of the cricoid), and glottic plane involvement for HSL (see Table 2). For some of these parameters, such as as true arytenoid fixation, even its ability to predict the pT classification alone is insufficient basing upon the computed kappa value. The role of head and neck exam with mirror and fiberoptic examination is prominent: in particular we believe it should be the only base for the assessment of laryngeal motility, which is eventually the main information for a safe recommendation of partial surgery as the probability of crico-arytenoid joint infiltration and of posterior paraglottic space invasion is proportional to the laryngeal motion impairment [40,41], and these are in turn the main factors impacting local failure in partial laryngeal surgery [17,32]. For other fundamental parameters, such as extensive cartilage invasion, the information comes exclusively from radiological evaluation. Anyway, for most of the parameters, our results confirm that the information from the two modalities is complementary, and their systematic integration has a decisive added value [22].

Among the limitations of the present study, there is the absence of MRI data. This is due to the difficulties in our institution to plan MRIs in due time, but it probably reflects the reality of most institutions worldwide and makes the present work a real-world study, which is, in our opinion, an added value. Furthermore, according to many authors, CT scan is usually preferred for laryngeal lesions because of a shorter scanning time, an excellent anatomic resolution, and a wide availability. Recently, emerging dual-energy CT techniques have been investigated for head and neck cancer imaging with the potential for improved tumor visualization and characterization [42].

In addition, LSCC patients may be non-compliant to performing MRI due to their difficulty in breathing and swallowing. In addition, the long acquisition time needed for MRI scans increases the risk of motion artifacts and consequently yields poor-quality images [43,44]. MRI is often reserved for selected cases, when CT results are not conclusive or there is an issue with CT or iodine contrast medium [43]. NCCN guidelines do not deem MRI study as mandatory in the work up of laryngeal cancer but consider it an alternative or an integration of CT scan information [45]. Our results show that no gross nor fatal mistakes have been made in treatment planning also without MRI.

However, in case of doubts concerning mostly the involvement of paraglottic or pre-epiglottic spaces, critical for the safe indications to surgical preservation [8,32,45], the MRI has been reported to bring an added value [46], but only with the availability of adequate technology and most of all of specifically trained skilled professionals. Without the availability of MRI and of specific skills and competence for exploiting its potential in laryngeal oncology, which is a frequent situation in the real-world setting, we feel comfortable in simply conservatively not indicating a partial surgery when there is a doubt concerning the above-mentioned features (most of all posterior lateral paraglottic space involvement). The oncological outcomes (in particular in terms of DSS) recorded in the present study confirm it is a wise attitude.

## Figures and Tables

**Figure 1 jpm-12-01585-f001:**
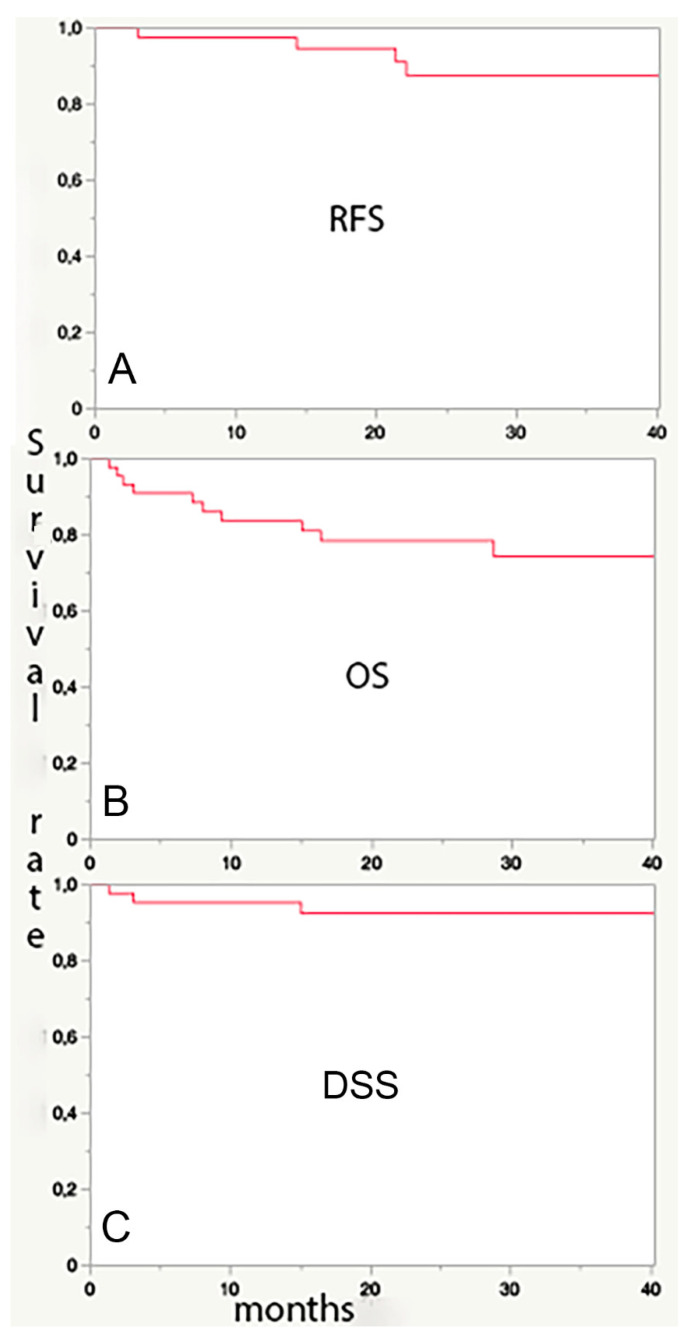
The very high relapse-free (panel (**A**), RFS: 88% at 3 years) and disease-specific survival (Panel (**C**), DSS: 92.5% at 3 years) rate in the present series of mostly advanced LSCC confirms the adequacy of the multidisciplinary work up. The markedly lower (74 % at 3 years) overall survival (OS, panel (**B**)) derives from the frequent comorbidities in these patients, usually heavy smokers and/or drinkers, which are a more common cause of death than LSCC itself.

**Figure 2 jpm-12-01585-f002:**
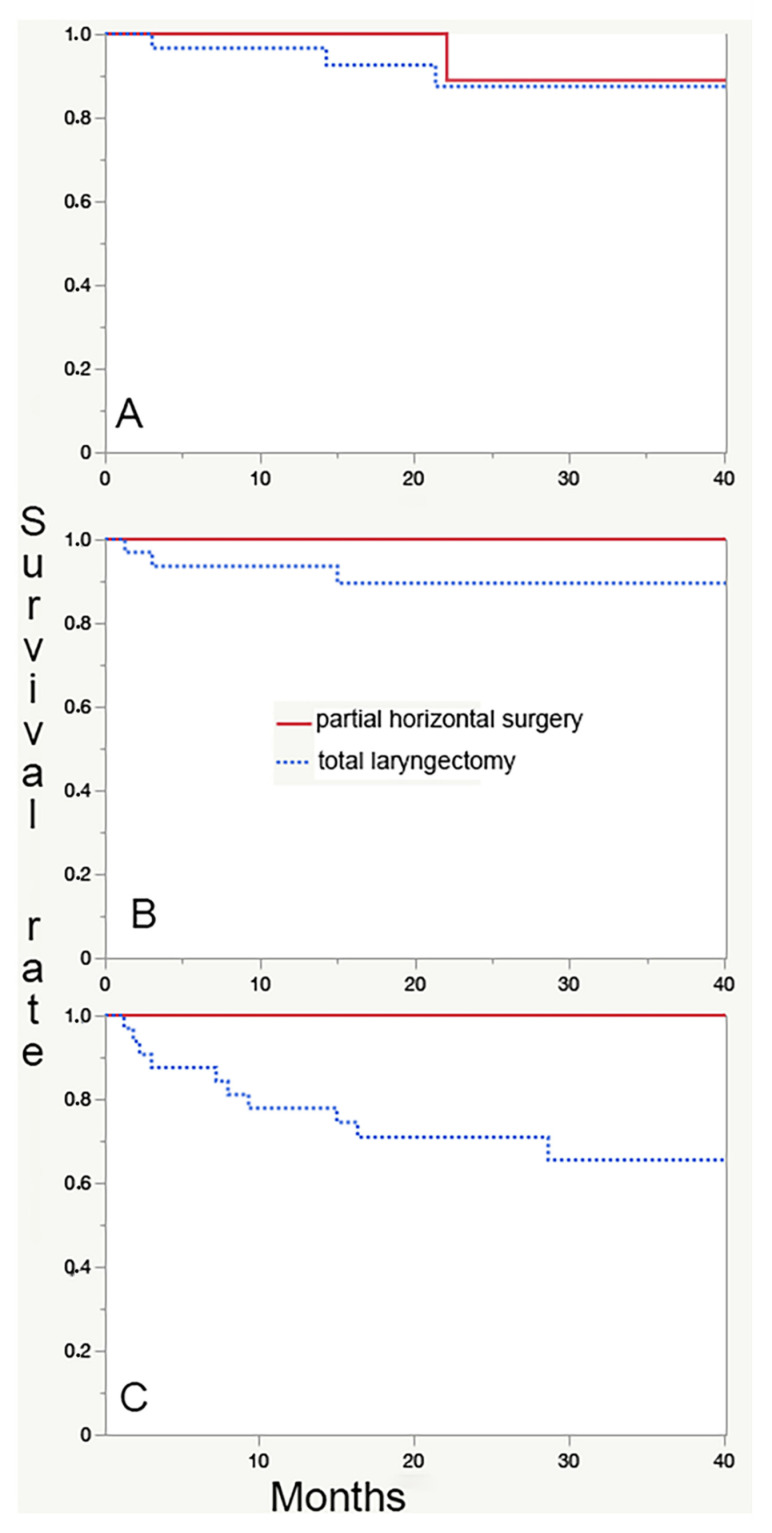
Comparison between survival curves in partial horizontal surgeries (*n* = 13, continuous line) versus total laryngectomies (*n* = 32, dotted line). No significant differences can be detected as for relapse-free (**A**) and disease-specific survival (**B**). Patients submitted to total laryngectomy have a significantly lower overall survival (**C**); this finding is expected because of the more advanced stage at diagnosis (and in particular nodal involvement) and the higher (in particular respiratory) comorbidity rate, which are both selection criteria for recommending a total laryngectomy versus a partial surgery.

**Table 1 jpm-12-01585-t001:** Descriptive statistics.

			*Surgical Treatment (n,%)*				
Parameters			Whole Series	Total laryngectomy 32 (71.1%)	Partial laryngeal surgery 13 (28.8%)		
					OPHL II 12 (26.6%)		EHSL 1 (2.2%)
Age (mean +/− SD)			65.7 +/− 10.6	65.9 +/− 10.6	64.3 +/− 8.3		75
Gender (*n*, %)							
Male			41 (91.1%)	29 (90.6%)	12 (100%)		0
Female			4 (8.9%)	3 (9.4%)	0		1 (100%)
Smoking status (*n*, %)							
Current			21 (46.7%)	16 (50%)	4 (34%)		0
Former			22 (48.9%)	16 (50%)	6 (50%)		1 (100%)
Never			2 (4.4%)	0	2 (16%)		0
Drinking status (*n*, %)							
Current			29 (64.5%)	19 (59.4%)	10 (84%)		0
Former			5 (11.1%)	5 (15.6%)	0		0
Never			11 (24.4%)	8 (25%)	2 (16%)		1 (100%)
Relevant comorbidities (*n*, %)							
yes			38 (84.4%)	25 (78.1%)	12 (100%)		1 (100%)
no			7 (15.6%)	7 (21.9%)	0		0
Renal failure 2 (5.2%)							
Liver failure 2 (5.2%)							
Diabetes 8 (21%)							
CPDO 4 (10.5%)							
Cardiovascular disease 11 (28.9%)							
Addiction 1 (2.6%)							
Other 10 (26.3%)							
Tumor site (*n*, %)							
Glottic larynx			29 (64.5%)	18 (56.3%)	11 (91.7%)		0
Supraglottic larynx			10 (22.2%)	8 (25%)	1 (8.3%)		1 (100%)
Hypopharynx			5 (11.1%)	5 (15.6%)	0		0
Subglottic larynx			1 (2.2%)	1 (3.1%)	0		0
Histology (*n*, %) SCC			45 (100%)	32 (100%)	12 (100%)		1 (100%)
Primary/Recurrence (*n*, %)							
Primary			41 (91.1%)	28 (87.5%)	12 (100%)		1 (100%)
Recurrence			4 (8.9%)	4 (12.5%)	0		0
Margins (*n*, %)							
R0			38 (84.4%)	26 (81.3%)	11 (91.7%)		1 (100%)
R1			1 (2.2%)	1 (3.1%)	0		0
Close			6 (13.4%)	5 (15.6%)	1 (8.3%)		0
Voice prosthesis (*n*, %)							
Yes			15 (33.3%)	15 (46.9%)	0		0
no			30 (66.7%)	17 (53.1%)	12 (100%)		1 (100%)
Pectoralis major flap (*n*, %)							
Yes			3 (6.7%)	3 (9.4%)	0		0
no			42 (93.3%)	29 (90.6%)	12 (100%)		1 (100%)
follow up +/− SD (months)			24.24 +/− 14.82	24.24 +/− 14.82	27.29 +/− 13.96		45.07
Recurrence (*n*, %)							
local			1 (2.2%)	1 (3.1%)	0		0
regional			2 (4.4%)	1 (3.1%)	1 (8.3%)		0
distant			2 (4.4%)	2 (6.3%)	0		0
no			40 (89%)	28 (87.5%)	11 (91.7%)		1 (100%)
Status (*n*, %)							
alive			35 (77.8%)	22 (68.7%)	12 (100%)		1 (100%)
dead			10 (22.2%)	10 (31.3%)	0		0
Cause of dead (*n*, %)							
tumor			3 (6.7%)	3 (9.4%)	0		0
other causes			7 (15.6%)	7 (21.9%)	0		0
cT							
T1			1 (2.2%)	0	0		1 (100%)
T1a			0	0	0		0
T1b			0	0	0		0
T2			14 (31.1%)	7 (21.9%)	7 (58.3%)		0
T3			25 (55.6%)	20 (62.5%)	5 (41.7%)		0
T4a			5 (11.1%)	5 (15.6%)	0		0
T4b			0	0	0		0
cN							
N0			34 (75.5%)	21 (65.6%)	12 (100%)		0
N1			8 (17.8%)	8 (25%)	0		0
N2a			0	0	0		0
N2b			3 (6.7%)	3 (9.4%)	0		0
N2c			0	0	0		0
N3a			0	0	0		0
N3b			0	0	0		0
pT							
T1			1 (2.2%)	0	0		1 (100%)
T2			13 (28.9%)	6 (18.7%)	7 (58.3%)		0
T3			21 (46.7%)	16 (50%)	5 (41.7%)		0
T4a			10 (22.2%)	10 (31.3%)	0		0
pN							
N0			33 (73.3%)	20 (62.5%)	12 (100%)		1 (100%)
N1			5 (11.1%)	5 (15.6%)	0		0
N2a			1 (2.2%)	1 (3.1%)	0		0
N2b			3 (6.7%)	3 (9.4%)	0		0
N2c			0	0	0		0
N3b			3 (6.7%)	3 (9.4%)	0		0
Clinical stage							
I			0	0	0		0
II			21 (46.7%)	8 (25%)	12 (100%)		1 (100%)
III			22 (48.9%)	22 (68.7%)	0		0
IVa			2 (4.4%)	2 (6.3%)	0		0
IVb			0	0	0		0
IVc			0	0	0		0
Comparison of T and N classifications							
ep TN		rad TN		c TN		p TN	
T *n*%	N *n*%	T *n*%	N *n*%	T *n*%	N *n*%	T *n*%	N *n*%
T1 1 (2.2%)	N0 40 (88.9%)	T1 1 (2.2%)	N0 34 (75.5%)	T1 1 (2.2%)	N0 34 (75.5%)	T1 1 (2.2%)	N0 33 (73.3%)
T2 22 (48.9%)	N1 4 (8.9%)	T1a 5 (11.1%)	N1 8 (17.8%)	T2 14 (31.1%)	N1 8 (17.8%)	T2 13 (28.9%)	N1 5 (11.1%)
T3 22 (48.9%)	N2b 1 (2.2%)	T1b 2 (4.5%)	N2b 3 (6.7%)	T3 25 (55.6%)	N2b 3 (6.7%)	T3 21 (46.7%)	N2a 1 (2.2%)
		T2 6 (13.3%)		T4a 5 (11.1%)		T4a 10 (22.2%)	N2b 3 (6.7%)
		T3 26 (57.8%)					N3b 3 (6.7%)
		T4a 5 (11.1%)					

**Table 2 jpm-12-01585-t002:** The analysis of reliability of the 3 different stagings obtained with endoscopy and physical exam (epTN), ceCT scans of the neck (chest CT was always performed, but M+ cases have not been included in the present work), (radTN), and with the integration of the first two order of data in the multidisciplinary discussion of the tumor board (cTN). Such reliability can be quantified with the concordance rate with the final histopathological exam (pT) as well as by the calculation of the Cohen kappa coefficient, which is an acknowledged measure of the interrater agreement and of the validity of a certain assay. Kappa values less than 0.4, between 0.4 and 0.75, or higher than 0.75 represent poor, fair-to-good, or excellent correlations, respectively. Kappa values of the cTN are always fair to good, cTN is better in the estimate of T classification than in that of N classification, even if the score improves if the simple assessment of nodal involvement and not the exact N classification is considered. When comparing the single parameters of the primary lesion as assessed by endoscopy and by ceCTscan, in comparison with the final histopathologic findings, a wide variation in the estimates can be appreciated, with better performance of endoscopy in some respects (superficial spread) and of morphological imaging in others (deep invasion). Some parameters, such as laryngeal mobility, can be properly assessed only with clinical assessment, and not even at final histopathological exam.

	ep TN	rad TN	c TN
T classification (agreement rate % with pT; kappa coefficient)	58%; 0.32	64%; 0.44	82%; 0.7
N classification (agreement rate % with pN; kappa coefficient)	76%; 0.27	76%; 0.42	76%; 0.42
N0 vs. N+ (agreement rate % with pN; kappa coefficient)	80%; 0.37	80%; 0.47	80%; 0.47
	**Single parameter (rate %)**		
	**Endoscopy/Physical exam**	**Radiology**	**Pathology**
Laryngeal vestibule/Piriform sinus inlet (Arytenoid/Aryepiglottic fold/suprahyoid epiglottis) involvement	35%	42%	37%
Lateral wall/apex of piriform sinus involvement	16%	20%	15%
Infrahyoid epiglottis involvement	26%	24%	23%
True vocal fold involvement	85%	80%	84%
Anterior commissure	77%	47%	60%
False vocal fold involvement	59%	62%	53%
Laryngeal mobility	Hypomobility 36% True vocal fold/arytenoid fixation 51%	-	-
Anterior paraglottic space involvement	8%	60%	58%
Posterior paraglottic space involvement	10%	27%	12%
Cricoid cartilage involvement	-	11%	11%
Thyroid cartilage involvement	-	24% *	21%
Subglottic extension	31%	18%	26%
Base of tongue involvement	8%	10%	10%

* Through cartilage invasion in 13%.

## Data Availability

Data are available upon request to the corresponding author.

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
