# Peer review of "Reliability of a Multidisciplinary Multiparametric Approach in the Surgical Planning of Laryngeal Squamous Cell Carcinomas: A Retrospective Observational Study"

_jpm, 2022, doi:10.3390/jpm12101585_

Round 1

Reviewer 1 Report

A very interesting article with solid results. A well designed study of great value. However, there is still room for improvement. First of all, partial laryngectomies are mainly indicated in salvage cases, after failure of conservative treatment. A paragraph should be devoted to the salvage laryngectomy and the results of it. These articles provide a lot of information and I think that it should be cited.

1)''Tsetsos N, Poutoglidis A, Vlachtsis K, Stavrakas M, Nikolaou A, Fyrmpas G. Twenty-year experience with salvage total laryngectomy: lessons learned. J Laryngol Otol. 2021 Aug;135(8):729-736. doi: 10.1017/S0022215121001687''

2) Silverman DA, Puram SV, Rocco JW, Old MO, Kang SY. Salvage laryngectomy following organ-preservation therapy - An evidence-based review. Oral Oncol. 2019 Jan;88:137-144. doi: 10.1016/j.oraloncology.2018.11.022

In addition PET/CT is referred as a scanning modality, but however its a very valuable imaging modalities, there are plenty pitfalls in oncology. Authors should refer in a sentence these limitation. This article provides information that can be employed and could be cited by the authors.

1) Tsetsos N, Poutoglidis A, Arsos G, Tsentemeidou A, Kilmpasanis A, Katsampoukas D, Fyrmpas G. 18F-FDG-PET/CT interpretation pitfalls in patients with head and neck cancer. Am J Otolaryngol. 2022 Jan-Feb;43(1):103209. doi: 10.1016/j.amjoto.2021.103209.

Author Response

Reviewer 1

A very interesting article with solid results. A well designed study of great value. However, there is still room for improvement. First of all, partial laryngectomies are mainly indicated in salvage cases, after failure of conservative treatment. A paragraph should be devoted to the salvage laryngectomy and the results of it. These articles provide a lot of information and I think that it should be cited.

1)''Tsetsos N, Poutoglidis A, Vlachtsis K, Stavrakas M, Nikolaou A, Fyrmpas G. Twenty-year experience with salvage total laryngectomy: lessons learned. J Laryngol Otol. 2021 Aug;135(8):729-736. doi: 10.1017/S0022215121001687''

2) Silverman DA, Puram SV, Rocco JW, Old MO, Kang SY. Salvage laryngectomy following organ-preservation therapy - An evidence-based review. Oral Oncol. 2019 Jan;88:137-144. doi: 10.1016/j.oraloncology.2018.11.022

Thank you very much for your precious suggestion, we added a paragraph in the introduction about salvage laryngectomies with the relative literature citations.

In addition PET/CT is referred as a scanning modality, but however its a very valuable imaging modalities, there are plenty pitfalls in oncology. Authors should refer in a sentence these limitation. This article provides information that can be employed and could be cited by the authors.

1) Tsetsos N, Poutoglidis A, Arsos G, Tsentemeidou A, Kilmpasanis A, Katsampoukas D, Fyrmpas G. 18F-FDG-PET/CT interpretation pitfalls in patients with head and neck cancer. Am J Otolaryngol. 2022 Jan-Feb;43(1):103209. doi: 10.1016/j.amjoto.2021.103209.

Thank you very much for your precious suggestion, we added a sentence in the discussion about the limitation of PET/CT reliability in head and neck oncology with the relative literature citations.

Reviewer 2 Report

It was an interesting article that discussed the importance of a multidisciplinary approach for LSCC. And the authors reported that The multidisciplinary approach is fundamental in the definition of the primary lesion in LSCC, in particular in order to safely perform surgical preservation of laryngeal function, which is associated with higher laryngectomy-free survival than irradiation but to a lower salvageability in case of recurrence. I agree that the importance of a multidisciplinary approach in clinical practice. But it seems less novelty and the case number was small in this study. In addition to the multidisciplinary approach, the article provides less information in their study

1.      Multidisciplinary management in tumor boards has become a standard in oncology, and has been demonstrated to significantly improve prognosis in head and neck oncology. In clinical practice, radiological staging and multidisciplinary discussion are the routines in cancer management. The authors should clearly describe the novelty of this article

2.      please check the item comorbidities in Table 1, and the enrolled patients were too small, it might be a limitation.

3. Please add AJCC staging in Table 1

4.  In addition to staging the disease, the benefits of a multidisciplinary approach should be presented and analyzed scientifically.

Author Response

Reviewer 2

It was an interesting article that discussed the importance of a multidisciplinary approach for LSCC. And the authors reported that The multidisciplinary approach is fundamental in the definition of the primary lesion in LSCC, in particular in order to safely perform surgical preservation of laryngeal function, which is associated with higher laryngectomy-free survival than irradiation but to a lower salvageability in case of recurrence. I agree that the importance of a multidisciplinary approach in clinical practice. But it seems less novelty and the case number was small in this study. In addition to the multidisciplinary approach, the article provides less information in their study

Thank you very much for your precious suggestion, we agree that the novelty of the present study was not fully described and outlined in the previous version. We do feel that the added value are

  • the “real world” setting, with only contrast enhanced CT scan and endoscopy available for staging, definition of the primary lesion and of N, where the real impact of multidisciplinary management can be fully appreciated;
  • the careful and extensive data collection and the relatively long follow up compensate for the relative smallness of the sample and allow us to obtain clear and interesting results for the readers of the journal;
  • the specific value of the multidisciplinary approach in the diagnostic work up and surgical planning of laryngeal cancer have not been previously specifically addressed.

We outlined the above aspects along the text in the new version of the manuscript.

  1.     Multidisciplinary management in tumor boards has become a standard in oncology, and has been demonstrated to significantly improve prognosis in head and neck oncology. In clinical practice, radiological staging and multidisciplinary discussion are the routines in cancer management. The authors should clearly describe the novelty of this article

Thank you very much for your observation, actually such aspect was not properly outlined in the first version of the manuscript, in the present version we outline in the beginning of the discussion that “even if the necessity of multidisciplinary management of head and neck cancers has been clearly and extensively proven, its utility for the specific aim of pretreatment definition of T classification and of anatomical pattern of spread of the primary lesion in LSCCs had been never previously evaluated”.

  1.     please check the item comorbidities in Table 1, and the enrolled patients were too small, it might be a limitation.

Thank you very much for the observations, you are right.

We checked the comorbidities as requested, there was a mistake in the no-comorbidity row, we are sorry and we amended it.

We also discussed in the text the relatively low number of patients which is a limitation but counterbalanced by the amount and detail of variables analyzed.

  1. Please add AJCC staging in Table 1

Thank you very much for your suggestion, AJCC staging has been added in the new version of Table 1.

  1. In addition to staging the disease, the benefits of a multidisciplinary approach should be presented and analyzed scientifically.

Thank you very much for your observation, you are clearly right, we tried to improve this aspect with many changes in the text:

  • We more systematically analyzed in the discussion the implications of differences in kappa values computed for the T and N presurgical classifications obtained (ep, rad, c) in comparison with pathological final report.
  • We also discussed more in detail the meaning of the different findings coming from the different evaluation modalities (endoscopy and radiology) of the primary lesion as summarized in table II, and the need of their systematic integration during the tumor board meetings.

Reviewer 3 Report

Dear authors,

this is a well presented study, with a varying interesting to readers, since one of the main drawback of the present study is the lack of novelty, as similar studies have been published. Nevertheless, it has a high significance with an important endpoint, so i personally believe that it is publishable worthy. I would like to stress out the main point that need some improvement is the article title, where it needs to be added the type of study, as a retrospective observational one. It could be also appropriate to present the work as an audit, as this is the most important conclusion of it. The rest of the STROBE guidelines should be checked prior to publication.

Author Response

Reviewer 3

this is a well presented study, with a varying interesting to readers, since one of the main drawback of the present study is the lack of novelty, as similar studies have been published. Nevertheless, it has a high significance with an important endpoint, so i personally believe that it is publishable worthy. I would like to stress out the main point that need some improvement is the article title, where it needs to be added the type of study, as a retrospective observational one. It could be also appropriate to present the work as an audit, as this is the most important conclusion of it. The rest of the STROBE guidelines should be checked prior to publication.

Thank you very much for your precious suggestion, we have outlined the main elements of novelty of the present work, in particular “even if the necessity of multidisciplinary management of head and neck cancers has been clearly and extensively proven, its utility for the specific aim of pretreatment definition of T classification and of anatomical pattern of spread of the primary lesion in LSCCs had been never previously evaluated”. We also modified the title as suggested and outlined in  materials and methods this is an audit. Passing through a further check of STROBE guidelines as suggested, we amended some missing information, as the definition of the present as a cohort study, and the description of the follow up and acquisition of relative data.

Round 2

Reviewer 2 Report

I think the revised manuscript could be accpeted